# Post-Traumatic Stress Disorder Symptoms among Pediatric Healthcare Workers

Nikolaos Rigas [1,*], Zacharias Kyritsis [2], Maria Dagla [1], Alexandra Soldatou [3], Eirini Orovou [4], Maria Tzitiridou-Chatzopoulou [4], Panagiotis Eskitzis [4] and Evangelia Antoniou [1]

1  Department of Midwifery, University of West Attica, 12243 Athens, Greece; mariadagla@uniwa.gr (M.D.); lilanton@uniwa.gr (E.A.)
2  Department of Mathematics, Aristotle University of Thessaloniki, 54124 Thessaloniki, Greece; zkyrit@math.auth.gr
3  Faculty of Medicine, National and Kapodistrian University of Athens, 15772 Athens, Greece; alsoldat@med.uoa.gr
4  Department of Midwifery, University of Western Macedonia, 50200 Ptolemaida, Greece; eorovou@uniwa.gr (E.O.); mtzitiridou@uowm.gr (M.T.-C.); peskitzis@uowm.gr (P.E.)
*  Correspondence: nrigas@uniwa.gr

**Abstract:** Post-traumatic stress disorder symptoms related to work in pediatric departments are major public health problems, as they directly affect health organizations, healthcare workers, and, due to the poor quality of care, the patients as well. The post-traumatic symptoms that a healthcare worker may experience are related to intrusion, avoidance, negative changes in cognition and mood, and changes in arousal and reactivity. The aim of the present investigation was to identify risk factors that contribute to the development of PTSD in pediatric healthcare workers, in order to implement necessary workplace measures. A sample of four hundred and forty-five pediatric workers at seven Greek public hospitals consented to participate in the survey. Socio-demographic data and a post-traumatic checklist (5th edition) were used to diagnose post-traumatic stress disorder. According to the results, risk factors for the development of post-traumatic symptoms include medical or nursing errors, threats to a child's life, and incidents of workplace bullying. More specifically, 25.2% of the participants had a provisional PTSD diagnosis, 72.8% of the sample experienced an incident involving a medical or nursing error in their workplace related to the treatment or care of a child, 56% experienced an incident involving a child's death or a threat to a child's life due to a serious illness or injury, and 55.5% experienced an incident of workplace bullying. In addition, it was found that having a university-level education, master's, or PhD, working in a circular shift, being assigned to a department by management rather than the worker, and dissatisfaction with salary were associated with post-traumatic stress disorder. The high rates of PTSD symptoms among participants highlight the need for prevention and management measures to protect and support the mental health of workers in pediatric departments. We propose frequent evaluations of the mental health of employees, more time for rest, incentives for professional development, utilization of their specializations and specialties, support from mental health specialists when symptoms are diagnosed, and the option for employees to change departments if they wish or if they show symptoms of mental trauma.

**Keywords:** pediatric departments; PICU; pediatric clinics; PTSD; healthcare workers; pediatricians; pediatric nurses

## 1. Introduction

Post-traumatic stress disorder (PTSD) symptoms related to work in hospital environments are major public health problems, as they directly affect health organizations, healthcare workers, and, due to the poor quality of care, the patients as well [1]. In particular, emergencies involving children are often considered to have a greater psychological impact on healthcare workers [2]. PTSD symptoms that a healthcare worker may have,

according to the Diagnostic and Statistical Manual (DSM-5), pertain to general symptoms of PTSD [3]. More specifically, the symptoms are classified into seven criteria: (criterion B) intrusion symptoms (unwanted upsetting memories, nightmares, and flashbacks); (criterion C) avoidance symptoms (avoidance of trauma-related stimuli); (criterion D) negative alterations in cognition and mood (feeling isolated, guilt or blame, inability to recall moments of trauma, negative thoughts, decreased interest in activities, and difficulty experiencing positive affect); (criterion E) alterations in arousal and reactivity (irritability or aggression, hyperactivity, heightened startle reaction, risky behavior, and difficulty sleeping); (criterion F) symptoms last more than one month; (criterion G) the symptoms must create significant functional impairment, social, professional, etc.; (criterion H) the symptoms must not be due to medication, substance use, or another illness. A prerequisite for the appearance of the above symptoms is that the individual has been exposed to a potentially stressful factor (criterion A), such as death, threatened death, actual or threatened serious injury, or actual or threatened sexual violence, in the following ways: (a) direct exposure, (b) witnessing the trauma, (c) learning that a relative or close friend has been exposed to trauma, or (d) indirect exposure to aversive details of the trauma, usually while performing professional duties (e.g., nurses, doctors) [4]. The prevalence of PTSD in healthcare workers in critical departments reaches 30%, much higher than the general population (3.5%) [2]. As emergencies involving children are often considered to have a greater psychological impact, pediatric healthcare workers may be at particular risk for PTSD [5]. According to the Centers forDisease Control and Prevention (CDC), over 80% of staff are emotionally and physically exhausted, while 45% of nurses reported not receiving enough emotional support [6]. In addition, staff often have to provide care (medical or nursing) without having processed stressful events or without having the opportunity to tend to their own care [7], given that pre-existing anxiety disorders and depression are risk factors for PTSD [8].

PTSD can seriously impact a healthcare worker's life, leading to an increased risk of severe fatigue, increased work errors, and even suicidal ideation or substance abuse [9,10]. Some of the reasons that bring about these problems are long hours, cyclical work, intense physical and emotional efforts, exposure to human suffering and death, and an increased risk of exposure to violence or disease [11,12]. On the other hand, research has shown that burnout is a significant predictor of PTSD symptoms [13,14].Burnout is characterized by emotional exhaustion, interpersonal cynicism, loss of personal identity, and ineffectiveness on both a personal and professional level; it is caused by exposure to chronic stress in the workplace [15,16] and usually occurs in workers with dangerous jobs that require high emotional involvement and responsibilities, such as nurses and doctors [17]. Previous studies have shown a bidirectional relationship between work-related anxiety and burnout [18,19]. Specifically, work-related anxiety increases the risk of violence in the workplace and, consequently, levels of burnout, while burnout, in turn, seems to worsen chronic work stress. As a result, daily work-related psychological injuries become more intense through the chronic stress of burnout, leading to the development of post-traumatic symptomatology. So, the higher the level of burnout, the more severe the symptoms of PTSD [20]. All of the above problems can be greater in pediatric care staff, as the loss of a child is an extremely tragic event, adding mental burden to health staff in departments treating pediatric patients [2,21,22]. However, staff in pediatric intensive care units (PICUs) and pediatric emergency departments face a greater risk of developing PTSD symptoms, due to the high rates of morbidity and mortality [2,7].

So far, several studies have evaluated the prevalence of PTSD, mainly in pediatric nurses [2,21–24], and a few in PICU staff [25–27]. Most of the above studies have focused on reactions and post-traumatic symptoms from occupational exposure to trauma; however, the process of symptom development is missing. As far as we know, there is no other study that has examined exposure injuries (criteria A) as risk factors in the development of PTSD. Considering that pediatric workers face a higher risk of developing PTSD, and with the aim of appropriately implementing measures and interventions, the purpose of this research study was to estimate the incidence of PTSD among healthcare workers in

pediatric departments and units, as well as to estimate the additional factors that contribute to the development of post-traumatic symptoms.

## 2. Materials and Methods

This cross-sectional study took place from October 2021 to June 2022, at pediatric departments, PICUs, and pediatric emergency departments of seven public university hospitals in Greece. These hospitals all had three pediatric specialties with a similar level of care. Ethics approvals from the study hospitals were obtained.

### 2.1. Participants

The survey participants were pediatric healthcare workers (doctors, nurses, and nursing assistants) from the above-mentioned university hospitals in Greece. A prerequisite for their participation was having worked for at least one year in the corresponding department [28]. Moreover, sufficient knowledge of the Greek language was another prerequisite for the healthcare workers, so they could understand the questions of psychometric tools. Participants were required to work full-time hours in the above departments and have a permanent employment relationship with the hospital. Of the seven university hospitals in the country (21 departments in total), 600 out of the 760 participants met the participation requirements for the study and agreed to receive and complete the questionnaires after being informed by the researcher. Out of these, 445 (75%) returned the completed questionnaires.

### 2.2. Measures

The data were collected after the researcher met with the health professionals, provided information, and obtained written consent for their participation in the research. This contact with the healthcare workers took place during a break from their work or afterward. The researcher from the University of West Attica was responsible for data collection.

The measures used for the needs of the research were as follows.

2.2.1. Socio-Demographic Questionnaire

A researcher-made questionnaire that included 21 items on personal, social, demographic, and occupational information.

2.2.2. Post-Traumatic Stress Checklist (PCL-5) [29]

This is a 20-item self-report scale that assesses PTSD symptoms from the last month, according to DSM 5 [30]. The PCL-5 was created by the staff of the National Center for PTSD. This version was translated and adapted to Greek by Orovou [31] in 2021 with very good psychometric properties (Cronbach $\alpha = 0.97$), and we used it after obtaining the necessary permission. The PCL-5 can be scored in two ways: (a) a score (range—0–80) can be obtained by summing the scores for each of the 20 items (higher scores indicate higher PTSD) and (b) for each group of criteria, any item that is rated 2 "Moderately" or higher is considered pathological. For example,1 or more pathological items of criterion B (questions 1–5), 1 or more pathological items of criterion C (questions 6–7), 2 or more pathological items of criterion D (questions 8–14), and 2 or more pathological items of criterion E (questions 15–20). In the present study, we used the second way to evaluate the groups of symptoms that exist in addition to the presence or absence of the disorder. This measure provides a provisional PTSD diagnosis.

### 2.3. Statistical Analysis

Statistical analysis was performed using SPSS software (version 27). Initially, descriptive statistics (mean value and standard deviation) were calculated for the post-traumatic stress criteria. In addition, the proportion (%) of pediatric workers experiencing PTSD based on the second diagnostic method of the PCL-5 instrument was recorded. Moreover, a forward analysis was carried out in order to highlight the relationship between

demographic factors, such as work-related aspects and other characteristics of pediatric healthcare workers, and the occurrence of post-traumatic stress. For this purpose, the Chi-square ($\chi^2$) independence test was used. Then, for the variables that emerged as important from the Chi-square independence test, logistic regression was performed to calculate the risk of post-traumatic stress regarding demographics, such as work-related aspects and other characteristics of pediatric healthcare workers. The significance level was set at $\alpha = 0.05$.

## 3. Results

Data from 445 pediatric healthcare workers were obtained and analyzed; there was a mean age of 41.3 years (SD = 10.1), mean work experience of 15 years (SD = 10.2), mean work experience in the specific department of 10.8 (SD = 9), and mean night shifts/on-calls per month of 5.4 (SD = 3.1). Other sample characteristics are presented in Table 1. Moreover, 385 of the participants (86.5%) were women and 60 (13.5%) were men. Most of the sample was nursing staff (65.4%), married or in a relationship (57.5%), working on a circular shift (68.1%), and had a university degree (38.9%).

**Table 1.** Demographics—occupational sample data.

|  |  | *n* | % |
|---|---|---|---|
| Gender | Man | 60 | 13.5% |
|  | Woman | 385 | 86.5% |
| Specialty | Medical staff | 154 | 34.6% |
|  | Nursing staff | 291 | 65.4% |
| Family status | Single | 160 | 36.0% |
|  | Married/In a relationship | 256 | 57.5% |
|  | Divorced | 29 | 6.5% |
| Education level | Secondary | 23 | 5.1% |
|  | University | 173 | 38.9% |
|  | Master | 123 | 27.6% |
|  | Ph.D. | 33 | 7.4% |
|  | Specialty/Expertise | 93 | 20.9% |
| Pediatric department | PICU | 155 | 34.8% |
|  | Emergencies | 50 | 11.2% |
|  | Clinic | 240 | 53.9% |
| Shift | Morning shift | 142 | 31.9% |
|  | Circular shift | 303 | 68.1% |

Table 2 shows the data regarding the department choice, the intention to change departments, and satisfaction with pay. The findings show that 55.3% (*n* = 246) of participants reported that the choice of their work department was not theirs. In addition, 24.5% (*n* = 109) of participants reported that—if given the opportunity—they would change their work department, while only 9.7% (*n* = 43) of participants reported that they were satisfied with their pay in relation to their work and the responsibility this entails.

**Table 2.** Data regarding department choice, intention to change departments, and satisfaction with salary.

|  |  | *n* | % |
|---|---|---|---|
| Choice to work in the specific department | Yours | 199 | 44.7% |
|  | Management | 246 | 55.3% |
| If you were given the opportunity, would you change departments? | Yes | 109 | 24.5% |
|  | No | 336 | 75.5% |
| Is your salary considered satisfactory in relation to your work and the responsibility it entails? | Yes | 43 | 9.7% |
|  | No | 402 | 90.3% |

The data presented in Table 3 show that, of the 445 healthcare workers, 72.8% (*n* = 324) had experienced an incident involving a medical or nursing error in their workplace, 56% (*n* = 249) had experienced an incident of death or a threat to a child's life, and 55.5% (*n* = 247) had experienced an incident involving workplace bullying.

**Table 3.** Traumatic events at work (criteria A).

|  | Yes | | No | |
|---|---|---|---|---|
|  | *n* | % | *n* | % |
| Nursing or medical error | 324 | 72.8% | 121 | 27.2% |
| Death or threat to the life of a child | 249 | 56.0% | 196 | 44.0% |
| Bullying | 247 | 55.5% | 198 | 44.5% |

*PTSD Levels in Pediatric Healthcare Workers*

Descriptive results for the PTSD scale questions that assess PTSD criteria are presented in Table 4. The results show that the mean value for the majority of PTSD scale items ranges between 0 (not at all) and 1 (a little), with some items showing an average value marginally above 1 (slightly). More generally, it appears that the symptomatology of PTSD among pediatric healthcare workers was at low levels.

**Table 4.** Descriptive analysis results for the 20-item PCL-5.

|  | M | SD |
|---|---|---|
| **Re-experiencing the traumatic event (criterion B)** | | |
| Are unwanted memories of the traumatic event recurring in a way that causes you to worry? | 0.9 | 1.1 |
| Are disturbing dreams related to the traumatic event recurring? | 0.8 | 1.0 |
| Do you feel and act as if the traumatic event is happening again and reliving it? | 0.8 | 1.0 |
| Do you get upset and agitated when something reminds you of a traumatic event of the past? | 1.2 | 1.1 |
| Do you feel like you have strange physical reactions, like a racing heart and sweating, every time you think about the traumatic event you experienced? | 1.0 | 1.1 |
| **Avoidance of situations reminiscent of the traumatic experience (criterion C)** | | |
| Do you avoid memories, thoughts, or feelings related to the specific traumatic event? | 1.3 | 1.3 |
| Do you avoid situations or activities that trigger memories and remind you of the traumatic event? | 1.1 | 1.2 |
| **Negative feelings about the critical event (criterion D)** | | |
| Do you have trouble remembering important elements of the traumatic event you experienced? | 0.7 | 0.9 |
| Do you have a negative belief about yourself, other people, and events happening in the world? (e.g., do you feel like everything is your fault, that you cannot trust anyone, or that the world is dangerous?) | 1.1 | 1.1 |
| Do you blame yourself or someone else for the traumatic experience, or what followed? | 0.8 | 1.1 |
| Do you feel strong negative emotions such as fear, terror, anger, guilt, or shame about the traumatic event you experienced? | 0.9 | 1.1 |
| Have you lost interest in activities you used to do in the past? | 1.1 | 1.1 |
| Do you feel distant or isolated from the people around you? | 0.9 | 1.1 |
| Do you feel emotionally apathetic or have difficulty expressing feelings of love to those close to you? | 0.8 | 1.0 |
| **Increased arousal and reactivity (Criterion E)** | | |
| Do you feel like you have irritability, outbursts of anger, or intense aggression? | 1.2 | 1.1 |
| Do you expose yourself to many risks or do things that can harm you? | 1.0 | 1.1 |
| Were you hypervigilant or alert (e.g., check to see who is around you, etc.)? | 0.8 | 1.4 |
| Do you feel startled or startled easily? | 1.0 | 1.1 |
| Do you find it difficult to concentrate on something? | 1.2 | 1.5 |
| Do you have trouble falling asleep or staying asleep? | 1.3 | 1.8 |

M = mean value, SD = standard deviation, 0 = not at all, 4 = very much.

Figure 1 shows the results regarding the percentage of workers in pediatric fields who experience PTSD. The results show that 25.2% (*n* = 112) of participants had characteristics that led to a diagnosis of PTSD.

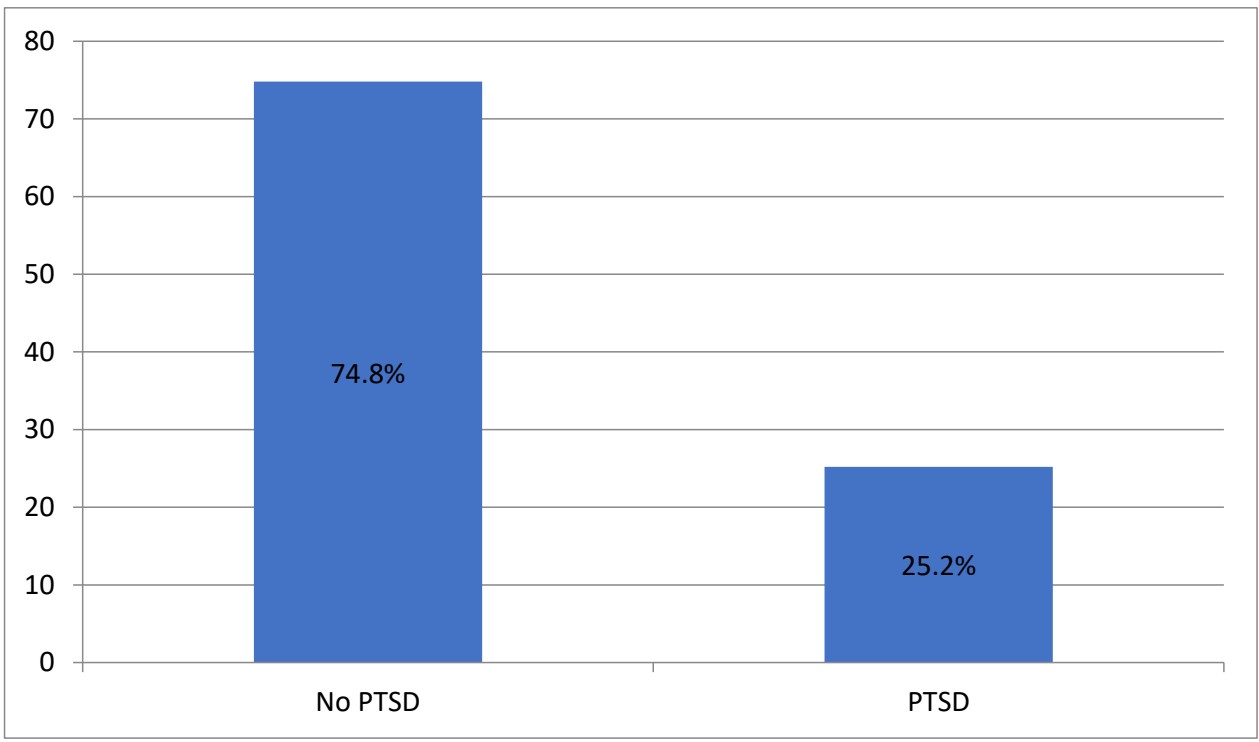

**Figure 1.** Percentage of PTSD of the participants.

The results show (Table 5) that 48.5% (*n* = 216) of participants met the conditions for PTSD criterion B, 44.3% (*n* = 197) of participants met the conditions for PTSD criterion C, 41.6% (*n* = 185) of participants met PTSD criterion D, and 44.5% (*n* = 198) of participants met PTSD criterion E.

**Table 5.** PTSD and criteria of PTSD results.

|  | **M** | **SD** | **Min** | **Max** | **%** |
|---|---|---|---|---|---|
| PTSD | 19.73 | 15.03 | 0.0 | 64.0 | 25.2% |
| Criterion B | 4.76 | 4.48 | 0.0 | 19.0 | 48.5% |
| Criterion C | 2.45 | 2.39 | 0.0 | 8.0 | 44.3% |
| Criterion D | 6.16 | 5.58 | 0.0 | 25.0 | 41.6% |
| Criterion E | 6.37 | 5.09 | 0.0 | 20.0 | 44.5% |

M = mean value, SD = standard deviation, Min = minimum value, Max = maximum value.

Table 6 shows that there is a statistically significant dependence between the occurrence of PTSD and the following characteristics of workers in pediatric fields: education level ($\chi^2(3)$ = 10.659, *p* = 0.014), working hours ($\chi^2(1)$ = 4.343, *p* = 0.037), choice of department ($\chi^2(1)$ = 4.030, *p* = 0.045), and satisfaction with salary ($\chi^2(1)$ = 10.506, *p* = 0.001). The analysis showed that, of the employees who were secondary education graduates, 4.3% (*n* = 1) were diagnosed with a provisional PTSD diagnosis. On the contrary, the percentages for university graduates, masters/PhD holders, and those in a specialty/specialization were 30.6% (*n* = 53), 24.4% (*n* = 38), and 21.5% (*n* = 20), respectively. In addition, the analysis showed that of the total number of employees who only worked the morning shift, 19% (*n* = 27) showed a provisional PTSD diagnosis, while the corresponding percentage for employees working in a circular shift was 28.1% (*n* = 85).

**Table 6.** PTSD diagnosis in relation to the demographics/work characteristics of workers in pediatric fields.

| | | Diagnosis | | | | $\chi^2$ | *p* |
|---|---|---|---|---|---|---|---|
| | | No PTSD | | PTSD | | | |
| | | *n* | % | *n* | % | | |
| Gender | Man | 44 | 73.3% | 16 | 26.7% | 0.083 | 0.774 |
| | Woman | 289 | 75.1% | 96 | 24.9% | | |
| Specialty | Medical staff | 113 | 73.4% | 41 | 26.6% | 0.265 | 0.607 |
| | Nursing staff | 220 | 75.6% | 71 | 24.4% | | |
| Education level | Secondary | 22 | 95.7% | 1 | 4.3% | 10.659 | 0.014 |
| | University | 120 | 69.4% | 53 | 30.6% | | |
| | Master/PhD | 118 | 75.6% | 38 | 24.4% | | |
| | Specialty/Expertise | 73 | 78.5% | 20 | 21.5% | | |
| Shift | Morning | 115 | 81.0% | 27 | 19.0% | 4.343 | 0.037 |
| | Circular | 218 | 71.9% | 85 | 28.1% | | |
| Department choice | Worker | 158 | 79.4% | 41 | 20.6% | 4.030 | 0.045 |
| | Management | 175 | 71.1% | 71 | 28.9% | | |
| Intention to change department | Yes | 74 | 67.9% | 35 | 32.1% | 3.566 | 0.059 |
| | No | 259 | 77.1% | 77 | 22.9% | | |
| Satisfaction with the salary | Yes | 40 | 93.0% | 3 | 7.0% | 10.506 | 0.001 |
| | No | 293 | 72.9% | 109 | 27.1% | | |
| Pediatric department | PICU | 116 | 74.8% | 39 | 25.2% | 0.858 | 0.651 |
| | Emergencies | 40 | 80.0% | 10 | 20.0% | | |
| | Clinic | 177 | 73.8% | 63 | 26.3% | | |
| Area | Region | 142 | 76.8% | 43 | 23.2% | 0.627 | 0.429 |
| | Athens | 191 | 73.5% | 69 | 26.5% | | |

Similarly, the findings showed that 20.6% (*n* = 41) of the total sample of employees who worked in a department of their own choice experienced PTSD, while the corresponding percentage for employees who worked in a department chosen by the management was 28.9% (*n* = 71). Finally, of all the employees who were satisfied with their salary, 7% (*n* = 3) showed PTSD, while the corresponding percentage for employees who were not satisfied with their pay was 27.1% (*n* = 109).

Table 7 shows the findings of the forward logistic regression analysis, with PTSD as the dependent variable and the demographics and work characteristics of the participants in the pediatric fields as independent variables, which emerged as statistically significant from the $\chi^2$ independence test.

**Table 7.** Statistically significant univariate logistic regression results.

| | B | Typical Error | d.f. | *p* | Exp(B) | 95% CI | |
|---|---|---|---|---|---|---|---|
| | | | | | | Lower | Upper |
| Shift | 0.507 | 0.249 | 1 | 0.042 | 1.661 | 1.019 | 2.706 |
| University | 2.274 | 1.036 | 1 | 0.028 | 9.717 | 1.276 | 73.977 |
| Master/PhD | 1.958 | 1.039 | 1 | 0.060 | 7.085 | 0.924 | 54.327 |
| Specialty/Expertise | 1.796 | 1.053 | 1 | 0.088 | 6.027 | 0.765 | 47.488 |
| Choice of department | 0.447 | 0.225 | 1 | 0.047 | 1.563 | 1.006 | 2.429 |
| Satisfaction with the salary | 1.601 | 0.609 | 1 | 0.009 | 4.960 | 1.503 | 16.365 |

Notes: d.f. = degree of freedom, reference groups: shift = morning, education = high school, department choice = yours, satisfaction with the salary = yes.

The analysis showed that participants in pediatric fields who worked in the circular shift were 1.661 times more likely to develop PTSD compared to those who worked in morning shifts (b = 0.507, *p* = 0.042, Exp(B) = 1.661, 95% CI = [1.019, 2.706]). Addition-

ally, pediatric workers working in non-self-selected departments were 1.563 times more likely to develop PTSD compared to those working in self-selected departments (b = 0.447, $p$ = 0.047, Exp(B) = 1.563, 95% CI = [1.006, 2.429]). Accordingly, pediatric workers who were not satisfied with their salary were 4960 times more likely to develop PTSD compared to those who were satisfied with their salary, (b = 1.601, $p$ = 0.009, Exp(B) = 4960, 95% CI = [1503, 16,365]). Finally, Table 8 shows that workers in pediatric fields, who were university graduates, were 9.717 times more likely to develop PTSD compared to those who were secondary education graduates (b = 2.274, $p$ = 0.028, Exp(B) = 9.717, 95% CI = [1.276, 73.977]).

**Table 8.** PTSD diagnosis in terms of the residual characteristics of participants.

| | | Diagnosis | | | | $\chi^2$ | $p$ |
|---|---|---|---|---|---|---|---|
| | | No PTSD | | PTSD | | | |
| | | *n* | % | *n* | % | | |
| Medical or nursing error | Yes | 227 | 70.1% | 97 | 29.9% | 15.928 | 0.000 |
| | No | 106 | 87.6% | 15 | 12.4% | | |
| Death or threat to the life of a child | Yes | 175 | 70.3% | 74 | 29.7% | 6.320 | 0.012 |
| | No | 158 | 80.6% | 38 | 19.4% | | |
| Bullying | Yes | 168 | 68.0% | 79 | 32.0% | 14.084 | 0.000 |
| | No | 165 | 83.3% | 33 | 16.7% | | |

Table 8 shows the results of the test of independence between the occurrence of PTSD and other characteristics of workers in pediatric fields. The results showed that there is a statistically significant dependence between the occurrence of PTSD and whether workers had experienced a medical or nursing error in their workplace ($\chi^2(1)$ = 15.928, $p$ = 0.000), had encountered a child's death or a life-threatening situation ($\chi^2(1)$ = 6.320, $p$ = 0.012), or had experienced any workplace bullying ($\chi^2(1)$ = 14.084, $p$ = 0.000).

Table 9 shows that workers who have experienced a medical or nursing error in their workplace exhibit PTSD at a rate of 29.9% (*n* = 97), while the corresponding rate for workers who have not experienced such an incident is12.4% (*n* = 15). In addition, the findings indicate that workers who have encountered a child's death or a life-threatening situation exhibit PTSD at a rate of 29.7% (*n* = 74), while the corresponding rate for workers who have not experienced such an incident is19.4% (*n* = 38). Finally, the results indicate that workers who have experienced workplace bullying developed PTSD at a rate of 32.0% (*n* = 79), while the corresponding percentage for workers who have not experienced workplace bullying is 16.7% (*n* = 33).

**Table 9.** Statistically significant results of forward logistic regression analysis, with the existence of PTSD as a dependent variable and independent/other work-related characteristics of workers in pediatric fields.

| | B | Typical Error | d.f. | $p$ | Exp(B) | 95% CI | |
|---|---|---|---|---|---|---|---|
| | | | | | | Lower | Upper |
| Medical or nursing error | 1.105 | 0.301 | 1 | <0.001 | 3.020 | 1.673 | 5.451 |
| Death or threat to the life of a child | 0.564 | 0.228 | 1 | 0.013 | 1.758 | 1.125 | 2.747 |
| Bulling | 0.855 | 0.234 | 1 | <0.001 | 2.351 | 1.485 | 3.723 |

Notes: d.f. = degree of freedom, reference groups = no.

Table 9 shows the findings of the forward logistic regression analysis, with the existence of PTSD as the dependent variable and the remaining characteristics of workers in pediatric fields as independent variables, which were shown to be statistically significant by the $\chi^2$ independence test. The analysis showed that workers in pediatric fields who

have experienced a medical or nursing error in their workplace were 3.020 times more likely to develop PTSD compared to those who have not experienced such an incident (b = 1.105, *p* = 0.000, Exp(B) = 3.020, 95% CI = [1.673, 5.415]). Furthermore, participants who had experienced the death of—or threat to the life of—a child were 1.758 times more likely to develop PTSD compared to those who had not experienced such an event (b = 0.564, *p* = 0.013, Exp(B) = 1.758, 95% CI = [1.125, 2.747]). Finally, participants who had experienced workplace bullying were 2.351 times more likely to experience PTSD compared to those who had not experienced workplace bullying (b = 0.855, *p* = 0.000, Exp(B) = 2.351, 95% D. E. = [1.485, 3.723]).

## 4. Discussion

The aim of the present investigation was to identify risk factors that contribute to the development of PTSD in pediatric healthcare workers in order to implement the necessary workplace measures. According to the results, 25.2% of participants were diagnosed with PTSD. In addition, 72.8%of the total sample had experienced a medical or nursing error in their workplace, 56% had experienced an incident involving a child's death or a threat to the life of a child, and 55.5% had experienced workplace bullying. Furthermore, it was found that PTSD in pediatric healthcare workers was associated with the circular shift, the department choice assigned by management (rather than by the worker), and dissatisfaction with salary.

There are many studies suggesting that daily exposure to psychological trauma involving children increases the risk of developing PTSD [2,32–39]. There are studies that have shown that medical or nursing errors have significant impacts on the mental health of healthcare workers. According to Wu [40], medical errors victimize healthcare workers, who require the same support as patients. Robertson [41] also argues that unintentional medical errors can have a lasting impact on certain workers, such as a lack of concentration, depression, burnout, and PTSD. In addition, Rassin's [42] research on nursing errors also aligns with our findings, as it showed that errors related to medication administration had a negative impact on the mental health of nurses. The results of this study also showed that the death (or threat of death) of a child is related to PTSD, and these findings are consistent with the findings of other studies. According to Bian's research [43], the care of terminally ill children is burdened with a plethora of negative emotions and anxiety, often resulting in post-traumatic symptomatology. Most nurses tend to approach death with an emotional focus, according to Kellogg [21], and this fact burdens their mental health. However, in a recent article [44], it was emphasized that factors that contribute to nurses' responses to grief may include previous experiences with death, personal traumatic experiences, attitudes toward nursing management, and the work environment. These factors contribute to the reactivation of past PTSD experiences. Another very important factor predicting PTSD in pediatric healthcare workers is the presence of workplace bullying, as indicated by our research findings. According to Chenevert [45], bullying is a factor that increases work-related stress, as it causes changes in the physical and mental conditions of the worker in response to life-threatening situations, such as challenges and threats. Additionally, Brande [46] defends that the consequences of workplace bullying are much more destructive than the consequences of all other occupational stressors combined. Nadal [47] also explains that the trauma of workplace bullying is similar to the trauma of domestic violence and sexual abuse, while Mikkelsen [48], in a study involving 118 victims of workplace bullying, found that the majority of participants who experienced bullying suffered from symptoms of re-experiencing, avoidance, hyperarousal, and functional impairment, supporting the idea that symptoms of PTSD area direct consequence of bullying.

Our results also showed that there is no statistically significant difference in PTSD rates among pediatric department workers (PICUs, emergencies, and clinics). Rocio Rodriguez-Rey [26] also found no differences in the PTSD rates between staff in different pediatric departments; the same conclusion was reached in an older research study by Mealer [10]

and Maytum [49]. On the contrary, the study by Secol [37] found that nurses in the pediatric oncology clinic had lower PTSD rates compared to nurses in the surgical clinic, even though child mortality rates in the former were clearly elevated. The findings of the study suggest that there are additional factors that contribute to the development of PTSD. For example, the higher educational level in our sample is associated with higher probabilities of PTSD. A possible explanation for this phenomenon involves the low job satisfaction of pediatric workers with a high level of education. Thus, the higher the educational level, the higher the individual's expectations, increasing the chances of job dissatisfaction [50]. Job dissatisfaction can also increase based on the healthcare worker's dissatisfaction with their salary [51]; this contributes to the development of PTSD [26,52] in trauma-exposed healthcare workers. Another factor that increases the chances of a trauma-exposed worker developing PTSD is cyclical shift, which, according to the literature, can lead to burnout syndrome [53]. The traditional eight-hour shifts for nurses may be a thing of the past for some countries; however, in Greece, they are still applied, usually distributed irregularly to ensure 24-h patient care [54]. Therefore, the insufficient quality and quantity of sleep, physical fatigue, chronic stress, and the generated burnout create the appropriate background for the development of PTSD [26,55,56]. In addition, the choice of the department by management rather than by the worker is related to the development of PTSD in our results. This phenomenon can be explained by the lack of job satisfaction and the burnout that results from the non-utilization of the specializations and special studies of pediatric workers. However, the PTSD rates we found (25.2%) were much higher than in the general population (3.5%) [2] and also higher than in other similar studies [2,32–39]. This situation can be explained by the high rates of burnout among doctors and nurses in Greece [57,58], as well as the lack of preventive measures and interventions for the mental health statuses of healthcare workers in Greece. Moreover, another important factor is the understaffing of the national health system with healthcare workers [58–60].

This study had strengths and limitations. A significant strength is the sufficient sample size that led to safe conclusions in the majority of cases. In addition, several variables were used to identify several factors that may lead to PTSD. The major limitation concerns the study design, which was cross-sectional, so we do not know the evolution of symptoms over time, as well as the factors that help to improve or worsen these symptoms. For example, through the follow-up of a prospective study, we could assess to what extent changing departments or receiving psychological support would affect the post-traumatic symptoms of the worker. Furthermore, due to the small number of secondary education graduates in our sample, we have some reservations about the results; therefore, we cannot draw conclusions about the participation of the educational level on PTSD.

## 5. Conclusions

This study identified several vulnerability factors that affect the mental injuries of pediatric workers in Greek public hospitals. The high rates of PTSD among the participants highlight the need to take measures to protect and support the mental health of workers in pediatric departments. We propose frequent evaluations of employees' mental health, more time for rest, incentives for professional development, utilization of their specializations and specialties, support from mental health specialists when symptoms are diagnosed, and the option for employees to change departments if they wish or if they show symptoms of mental trauma. Another important measure is to increase the number of healthcare professionals, which should be seriously considered by decision-makers as part of the solution to the problem. Furthermore, targeted studies should evaluate additional PTSD development factors, such as the role of burnout, but also evaluate interventions aimed at increasing psychological resilience and reducing burnout symptoms, which are prognostic factors of PTSD in pediatric workers.

**Author Contributions:** Conceptualization, N.R.; methodology, N.R. and Z.K.; software, Z.K.; validation, A.S., M.D. and E.A.; formal analysis, N.R. and Z.K.; investigation, N.R.; resources, M.T.-C.; data curation, N.R., Z.K. and P.E.; writing—original draft preparation, N.R.; writing—review and editing,

E.A.; visualization, E.O.; supervision, E.A. All authors have read and agreed to the published version of the manuscript.

**Funding:** This research received no external funding.

**Institutional Review Board Statement:** The study was conducted in accordance with the Declaration of Helsinki, and approved by the (1) Children's Hospital Agia Sofia in Athens Ethics Commission: 14972/30-07/2021, (2) "Pan. & Aglaia Kyriakou" Children's Hospital in Athens Ethics Commission: 12652/14-07-2021, (3) Children's Hospital in Athens, Attikon University General Hospital in Athens Ethics Commission: 6/7-7-2021, (4) General Children Hospital of Penteli in Athens Ethics Commission: 6425/25-06-2021, (5) General Hospital Hippokration, in Thessaloniki Ethics Commission: 23566/23-09-2021, (6) General University Hospital of Patras Ethics Commission: 330/06-07-2021, (7) University General Hospital of Heraklion "PAGNI" in Crete Ethics Commission: 16246/21-9-21.

**Informed Consent Statement:** Informed consent was obtained from all subjects involved in the study.

**Data Availability Statement:** The data used to support the findings of this study are available from the corresponding author upon request.

**Public Involvement Statement:** There was no public involvement in any aspect of this research.

**Guidelines and Standards Statement:** This manuscript was drafted against the STROBEcriteria for reporting cross-sectional studies

**Conflicts of Interest:** The authors declare no conflict of interest.

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
