# Peer review of "Post-Traumatic Stress Disorder Symptoms among Pediatric Healthcare Workers"

_nursrep, doi:10.3390/nursrep14010010_

Round 1
Reviewer 1 Report
Comments and Suggestions for Authors
In this manuscript, the authors present an evaluation of Post-traumatic Stress Disorder Symptoms among Pediatric Health Care Workers.
The work is relevant, and the manuscript is well-written and organized. The authors interviewed 445 pediatric healthcare workers in seven hospitals.
In the introduction, you say the prevalence of PTSD in the general population is 3.5%. In the Discussion, you say 6%. Which one is correct?
How do you differentiate job dissatisfaction from PTSD? I suppose they are related.
There are several typos in the manuscript. Here, I will show a few of them, but the manuscript needs a review.
-mental damage
-the symptomatology of PTSD among pediatric health care workers in fields is at low levels
-Umiversity
-72.8%of
-founds that found that
-PTSDdiagnosisin terms
Comments on the Quality of English Language
There are several typos in the manuscript. Here, I will show a few of them, but the manuscript needs a review.
-mental damage
-the symptomatology of PTSD among pediatric health care workers in fields is at low levels
-Umiversity
-72.8%of
-founds that found that
-PTSDdiagnosisin terms
Author Response
Dear Reviewer
Thank you for your support.
I wish you a Happy New Year!
In this manuscript, the authors present an evaluation of Post-traumatic Stress Disorder Symptoms among Pediatric Health Care Workers.
The work is relevant, and the manuscript is well-written and organized. The authors interviewed 445 pediatric healthcare workers in seven hospitals.
In the introduction, you say the prevalence of PTSD in the general population is 3.5%. In the Discussion, you say 6%. Which one is correct?
it was a mistake we fixed it. Thank you!
How do you differentiate job dissatisfaction from PTSD? I suppose they are related.
PTSD is a psychiatric disorder that can develop after someone experiences or witnesses a traumatic event. It is characterized by symptoms such as flashbacks, nightmares, anxiety, and difficulty sleeping. Work dissatisfaction is considered to be a factor that can help in the development of PTSD.
There are several typos in the manuscript. Here, I will show a few of them, but the manuscript needs a review.
-mental damage
-the symptomatology of PTSD among pediatric health care workers in fields is at low levels
-Umiversity
-72.8%of
-founds that found that
-PTSDdiagnosisin terms
Thank you very much! We have corrected them.
Yours sincerely
Nikolaos Rigas
Reviewer 2 Report
Comments and Suggestions for Authors
Thank you for the opportunity to review this manuscript. The paper is overall well written and well presented.
1. What are the objectives and Hypotheses of this study? It is necessary to elaborate on the hypotheses and present the related results.
1. It needs to explain the practical implications of this study in a more detailed way.
Author Response
Dear Reviewer
Thank you for your support. We have corrected them!
I wish you a Happy New Year!
Nikolaos Rigas
Reviewer 3 Report
Comments and Suggestions for Authors
Thank you for submitting the manuscript, entitled “Post-traumatic Stress Disorder Symptoms among Pediatric Health Care Workers” to Nursing Reports. You can see my comments below.
Title:
· What is the cause of the PTSD the authors would like to investigate?
Abstract
· The authors should check the punctuations following the Journal instructions.
· Rewrite this sentence, “The symptoms that a health care worker may experience, refer to the general symptoms of posttraumatic stress disorder;”
· What do you mean (5 edition)?
· What do you mean “necessary work measures”? it’s unclear.
· 25.2% of the participants were diagnosed with PTSD by doctors or using the survey? If using the survey, the PTSD survey can serve as a tool to detect the possible or risk of PTSD. Please clarify.
· The results are not clear. The incidence of medical or nursing errors and incident of death… were the factors? Please clarify and rewrite this part.
Introduction
· The literature review and knowledge gap are not clear.
· Some factors were mentioned in the Introduction but it is still inadequate to explain the necessity of your study. What have been known and what is still unknown? The authors should elaborate more why the study is important.
Method
· Line 92 “7” should be written “seven”
· Lines 93 to 97, it’s not necessary to list out the departments but highlight all were in paediatric specialties.
· Lines 98 to 105, it’s not necessary to list out all ethics approvals but highlight that the ethics approvals from study hospitals were obtained. The list-out information should be added in the supplementary section at the end of paper.
· By the way, what about the nature of all study venues. Are they at the similar level of healthcare services? Acute, sub-acute, or rehabilitative?
Participants
· Any inclusion and exclusion criteria related t the job nature?
· How about sample size calculation?
· Considering involvement of a number of hospitals/pediatric units, how to determine the number of participants in each venue?
Measures
· Who was responsible for the data collection?
· The demographics included 21 themes or items?
· Regarding the PCL-5, how was it interpreted? Higher scores indicate higher PTSD? Elaborate the scoring system.
Analysis
· What method of logistic regression was used, such as forward, backward, or stepwise?
· Chi-square always, not chi
· Check spelling error
Results
· The results should be interpreted with caution because some factors, such as educational level. There was a relatively small sample size in the participants with higher education that may not accurately to confirm its effects on PTSD; same to the demographic group in small sizes.
· In Table 5, what about Criterion A?
· Use past tense in this section
· In Table 10. P=.000 should be written as p<0.001
· The discussion seems like another introduction or background. The authors should rewrite this part and interpret the results. The authors should make well use of the results to interpret and make impacts of the current practice, especially to help healthcare worker to prevent/reduce the PTSD. However, the authors made very superficial explanation of the results and then jumped to the practice. If the results are not well interpreted, the implication of the practice is not evidence-based enough.
others
· Please check the format of referencing including in-text citations that cannot meet the Journal requirement.
Comments on the Quality of English Language
· Need English proficiency check
Author Response
Dear Reviewer
Thank you very much for allowing me to revise the article. I have responded to all your comments and made all the revisions to the text.
Warmest Wishes for a Happy New Year!
Title:
- What is the cause of the PTSD the authors would like to investigate?
The prerequisite for the symptoms of PTSD to appear is previous exposure to a traumatic event that is life-threatening, through various types of exposure, such as direct exposure, exposure as a witness, third-party exposure, and daily exposure to traumatic events through work.
This last form of exposure refers to the daily contact of healthcare professionals with human suffering, serious injuries, and death. In this study, the cause of posttraumatic symptomatology is the daily contact of the staff with the diagnosis and care of a child with a serious illness, the child's serious injury, and death.
Abstract
- The authors should check the punctuations following the Journal instructions.
Thank you for your comment. The abstract has been revised according to the journal instructions.
- Rewrite this sentence, “The symptoms that a health care worker may experience, refer to the general symptoms of posttraumatic stress disorder;”
This sentence was rewritten precisely clarifying the post-traumatic symptomatology.
- What do you mean (5 edition)?
It is the 5th and last version of the DSM-5 that was released in 2013. "DSM-5" is the official name of the fifth edition of the Diagnostic and Statistical Manual of Mental Disorders.
- What do you mean “necessary work measures”? it’s unclear.
It has been corrected. The text stated that preventive measures and measures to address the mental disorders of the personnel need to be taken.
- 25.2% of the participants were diagnosed with PTSD by doctors or using the survey? If using the survey, the PTSD survey can serve as a tool to detect the possible or risk of PTSD. Please clarify.
Thank you very much for this comment. The evaluation was conducted using a specific psychometric tool the PCL-5, which provides a provisional diagnosis of PTSD. Therefore, we do not have complete and definitive diagnoses that come from a psychiatric examination. It has been corrected everywhere in the text. The use of the scale is done for research purposes but also for screening or before diagnosis in population groups.
- The results are not clear. The incidence of medical or nursing errors and incident of death… were the factors? Please clarify and rewrite this part.
Thank you for the comment, the point has been corrected. The factors you mentioned are risk factors for the development of PTSD, as the workers exposed to them.
Introduction
- The literature review and knowledge gap are not clear.
- Some factors were mentioned in the Introduction but it is still inadequate to explain the necessity of your study. What have been known and what is still unknown? The authors should elaborate more why the study is important.
Thank you very much for this intervention. The importance of conducting this study was indeed missing in this article, which we have added.
Method
- Line 92 “7” should be written “seven”
Corrected
- Lines 93 to 97, it’s not necessary to list out the departments but highlight all were in paediatric specialties.
They have been removed as you requested.
- Lines 98 to 105, it’s not necessary to list out all ethics approvals but highlight that the ethics approvals from study hospitals were obtained. The list-out information should be added in the supplementary section at the end of paper.
This paragraph was removed after your recommendation.
- By the way, what about the nature of all study venues. Are they at the similar level of healthcare services? Acute, sub-acute, or rehabilitative?
Thank you very much for pointing that out. It has been clarified that it was the same level.
Participants
- Any inclusion and exclusion criteria related t the job nature?
“Survey participants were all pediatric health care workers (doctors, nurses and nursing assistants), from the above University hospitals in Greece. The participants participated provided that they had worked for at least one year in the corresponding department [20]. Moreover, sufficient knowledge of the Greek language was another condition for the health care workers so they could understand the questions of psychometric tools”.
In addition to what was already written, the following sentence was added “Participants also were required to work full-time hours in the above departments and have a permanent employment relationship with the Hospital”.
- How about sample size calculation?
- Considering involvement of a number of hospitals/pediatric units, how to determine the number of participants in each venue?
Thank you very much for this question. We have added the number of the initial sample and the final sample of our study in the "participants" section.
I am also given the opportunity to mention the significant problem of understaffing in Greece's national healthcare system, both in terms of doctors and nurses. There are 1-2 nurses per shift, while in intensive care units, one nurse usually cares for 4 or sometimes even 5 ICU beds.
As a result of an understaffing, the personnel (doctors, nurses) can be shared between a nursing department and an emergency department for the aforementioned reasons.
Measures
- Who was responsible for the data collection?
The researcher was responsible for data collection. It was added to the section "measures”.
- The demographics included 21 themes or items?
The demographics included 21 items. It was corrected.
- Regarding the PCL-5, how was it interpreted? Higher scores indicate higher PTSD? Elaborate the scoring system.
Thank you very much for giving me the opportunity to clarify and make the scale PCL-5 more understandable.
We used the second method, therefore, for each group of criteria, any item that is rated with 2 "Moderately" or higher, is considered pathological. For example, 1 or more pathological items of criterion B (questionsitems 1–5), 1 or more pathological items of criterion C (questionsitems 6–7), 2 or more pathological items of criterion D (ques-tions 8-14 ), and 2 or more pathological items of criterion E (questions 15-20).
Analysis
- What method of logistic regression was used, such as forward, backward, or stepwise?
Forward logistic regression was used. We have fixed it everywhere. Thank you!
- Chi-square always, not chi
I did it. Thank you.
- Check spelling error
I did it. Thank you!
Results
- The results should be interpreted with caution because some factors, such as educational level. There was a relatively small sample size in the participants with higher education that may not accurately to confirm its effects on PTSD; same to the demographic group in small sizes.
The PhD holders are 33 (7.4%) and secondary school graduates are 23 (5.1%). In most studies, those with a PhD and those with a low educational level are included. Participants with University education are 38.9%.
It was an error (higher educational level, we meant secondary educational level). It was corrected. Additionally, the small number of high school graduates has been added to the study limitations.
- In Table 5, what about Criterion A?
The PCL-5 scale includes only criteria B, C, D, and E. The remaining criteria (G, F, H), as well as criterion A, are necessary prerequisites. Specifically, criterion A is the traumatic event that someone is exposed to. In our research, criteria A were identified as Nursing or Medical error, Death or threat to the life of a child and Bullying.
- Use past tense in this section
Thank you reviewer, you help me a lot. I corrected them.
- In Table 10. P=.000 should be written as p<0.001
I corrected it
The discussion seems like another introduction or background. The authors should rewrite this part and interpret the results. The authors should make well use of the results to interpret and make impacts of the current practice, especially to help healthcare worker to prevent/reduce the PTSD. However, the authors made very superficial explanation of the results and then jumped to the practice. If the results are not well interpreted, the implication of the practice is not evidence-based enough.
Thank you very much for this support. The discussion has been revised as you will see almost entirely.
others
- Please check the format of referencing including in-text citations that cannot meet the Journal requirement.
I did it. Thank you.
Kind Regards
Nikolaos Rigas
Round 2
Reviewer 3 Report
Comments and Suggestions for Authors
Thank you for the revision. The manuscript is well revised. The authors should review the paragraphs and combine if appropriate. It's not suitable to have one single sentence to be a paragraph.
Please check reference 22 which seems that it's incomplete.
Author Response
Dear Editor
Thank you very much on behalf of all the co-authors for your valuable help in the article. I have made the necessary corrections you requested and highlighted them in green.
Yours sincerely
Nikolaos Rigas